



# Anatomy of simultaneous flood peaks at a lowland confluence

Geertsema Tjitske J.[1], Teuling Adriaan J.[1], Uijlenhoet Remko[1], Torfs Paul J.J.F.[1], and Hoitink Ton J.F.[1]

[1]Hydrology and Quantitative Water Management Group, Wageningen University, Wageningen, The Netherlands.

**Correspondence:** Tjitske Geertsema (tjitske.geertsema@wur.nl)

**Abstract.** Lowlands are vulnerable to flooding due to their mild topography in often densely populated areas with high social and economic value. Moreover, multiple physical processes coincide in lowland areas, such as those involved in river-sea interactions and in merging of rivers. Simultaneous occurrence of such processes can result in amplifying or attenuating effects on water levels. Our aim is to understand the mechanisms behind simultaneous occurrence of discharge waves in a river and its lowland tributaries. Here, we introduce a new way of analyzing lowland discharge and water level dynamics, by tracing individual flood waves based on dynamic time warping. We take the confluence of the Meuse river ($\sim$33,000 km$^2$) with the joining tributaries of the Dommel and Aa rivers as an example, especially because the January 1995 flood at this confluence was the result of the simultaneous occurrence of discharge peaks in the main stream and the tributaries. The analysis shows that the exact timing of the arrival of discharge peaks is of little relevance, because of the long duration of the average discharge wave compared to typical time lags between peaks. The discharge waves last on average 9 days, whereas the lag time between discharge peaks in the main river and the tributaries is typically 3 days. This result in backwaters that can rise up to 1.5 m over a distance of 4 km from the confluence.

## 1 Introduction

In January 1995, water was only a few centimetres below the crest of the dikes in the lowlands of the major rivers Rhine and Meuse. This high water event created fears for major flooding across large parts of the Netherlands, causing the Dutch government to decide to evacuate more than 200,000 inhabitants from the area immediately under threat. Luckily these fears did not materialise and people could return to their homes within a few days. However, this high water event did lead to local flooding, for instance near the city of 's Hertogenbosch, where a major European highway was inundated (Figure 1). The flood occurred just upstream of the city center, where two tributaries (Dommel and Aa) join, and spread over a small region. This small region included an economically important highway and could have spread over a much larger area if the duration of the overtopping would have been longer (Figure 1). The overtopping was the result of water levels rising to 4.9 m above msl in the Dommel, indicated by the water authorities as maximum protection level. It was concluded that the simultaneous occurrence of the discharge peaks in the Meuse river and the Dommel tributary likely caused the flooding, but no in-depth analysis of the simultaneous occurrence has been performed so far.

It can be argued that the coincidence of discharge peaks from the main river and a lowland tributary is unlikely to occur. Firstly, the spatially averaged precipitation over a larger river basin and the lowland tributary might show little correlation





(Betterle et al., 2017). Precipitation events exceeding 1 day can cover more than $10^4$ km$^2$ (Merz and Blöschl, 2003; Skøien et al., 2003; Van de Beek et al., 2011, 2012) and may therefore be comparable for a part of the river basin, but not for the entire 33,000 km$^2$ basin. Secondly, even if the precipitation pattern would be homogeneous across both catchment areas, then the shorter drainage network length of the lowland tributary should result in discharge peaks reaching the confluence before

the peak in the main river (Marchi et al., 2010; Melone et al., 2002). The Meuse is a rain-fed river receiving input from multiple lowland tributaries with much shorter network lengths (Berger, 1992; De Wit et al., 2007). For example, the response times from rainfall to outlet of the Dommel and Aa tributaries are about 2 days based on a prediction from the corresponding catchment areas (formulae of Mitchell (1948) in Melone et al., 2002), whereas the estimated response time of the Meuse river is around 13 days. Such a difference in response times suggests that simultaneous occurrence of discharge peaks should not

be an issue at the Dommel and Meuse confluence, provided that both peaks are triggered by the same system. Therefore, we postulate the hypothesis that simultaneous occurrence of discharge peaks is unlikely at lowland confluences of medium-sized rivers (river lengths between 300 and 1000 km). The event described above seems to contrast with this hypothesis, but it is unclear whether the 1995 event reflects a general pattern of possible simultaneous occurrence of flood peaks or simply a rare extreme event.

While simultaneous occurrence (coincidence) of high discharge and storm surge has been investigated (Kew et al., 2013; Klerk et al., 2015; Van Den Hurk et al., 2015), only few studies have addressed the processes leading to simultaneous occurrence of discharge peaks. In one of these studies, Pattison et al. (2014) analyzed the sensitivity of changing hydrological response to the relative timing and succession of discharge peaks by sub-watersheds, to investigate the possibility of reducing flood risk. The cause of the relative timing and succession, however, was not considered in their analysis, making it difficult

to extrapolate their findings. A key point from their analysis was that flood impacts are unpredictable when synergy occurs between two sub-watersheds. In another study by Vorogushyn and Merz (2013), it is shown that a discharge wave accelerates with river training, resulting in simultaneous occurrence of flood peaks in the Rhine river and the Neckar tributary, which partly causes increasing Rhine flood discharges. Vorogushyn and Merz (2013) analysed the simultaneous occurrence of discharge peaks of the Rhine and Neckar based on trend analysis, without offering details of individual discharge events. Here,

we demonstrate that the variability in the hydrograph shapes of individual peak discharge events is so large, that changes in the relative timing cannot readily be translated to a change in flood risk.

Peak discharge hydrographs vary in time due to heterogeneity in precipitation patterns and antecedent catchment characteristics. A single flood event may be insufficiently representative to gain generic insight into the conditions when discharge peaks coincide and what the consequences are. Here we carry out an analysis of a selection of events leading to the highest

discharges, and establish the impacts on water levels. The precipitation patterns for these events provide insight in the variety of conditions that may lead to simultaneous occurrence of discharge peaks, whereas water level analysis offers insight into the possible consequences. Obviously, extreme water levels in tributaries can lead to inundation of larger areas when the surrounding region is flat (i.e. in lowlands). This study aims to increase our understanding of simultaneous flood peak occurrence at confluences in lowland areas, by investigating the precipitation, discharge and water level dynamics for peak discharge events in the lower branch of the River Meuse. We will address the following research questions:





- Under which conditions do discharge peaks coincide at the selected lowland confluences?

- How do water level profiles in lowland tributaries respond to simultaneous occurrence of discharge peaks?

The lowland Dommel and Meuse confluence is used as study area for two reasons. First, because simultaneous occurrence
reportedly occurred in January 1995. Second, the study area is uniquely suited for this research since routine and independent
observations of water height and discharge are available at several locations in the Meuse and its tributaries. The nine highest
discharge events of the Meuse between 1999 and 2015 are analyzed in detail, to gain insight in the spatio-temporal precipitation
and discharge patterns. For these events, the time lags between peaks in the Meuse and the Dommel and Aa are determined,
as the timing indicates the potential of simultaneous occurrence of discharge peaks. We employ a method that is relatively new
in hydrology to calculate time lags, dynamic time warping (DTW), which is introduced in Section 2 (Methods and Materials).
The Meuse river basin, its tributaries and the employed discharge and precipitation data are described in more detail in the
same section. Section 3 (Results) shows the precipitation patterns prior to these discharge events, and the resulting time lags
between the Meuse and the tributaries Dommel and Aa for the studied discharge peaks. In addition, the effects of simultaneous
occurrence of discharge peaks on water levels in the rivers are analyzed. Finally, we will discuss our results and conclude that
the exact timing of the discharge peaks is not the single and most important factor in explaining the hydrological consequences
of the simultaneous occurrence of discharge peaks at lowland confluences (Section 4, Discussion, and Section 5, Conclusions).

## 2 Methods and Materials

### 2.1 Dynamic Time Warping

Dynamic time warping (DTW) is a relatively new method and has so far seen few applications in the field of hydrology
(Ouyang et al., 2010; Dupas et al., 2015). The advantage of DTW is that no assumptions are needed regarding the wave traced
in discharge time series. This in contrast to common methods based on the centre of mass and the unit hydrograph. DTW is
unique in considering the time axis elastic / dynamic, which is desirable to match similar shapes in different phases, such as
long wave propagation at confluences Kruskal and Liberman (1983); Keogh and Ratanamahatana (2005) provide a detailed
description of the DTW method. Here, we describe the essence of the method using two standardized discharge time series as
an example (Figure 4), $x_i$ (upstream) with $i = 1..n$ and $y_j$ (downstream) with $j = 1..m$. The Euclidean distance, $d(x_i, y_j)$,
between the two time series is expressed as an $n$-by-$m$ matrix,

$$d(x_i, y_j) = |x_i - y_j|, \tag{1}$$

which should not be confused with the physical distance between the two stations. The distance is used to find optimal matches
by minimizing the distances. For optimal matching of the minimal distances, the amplitudes of x and y should be similar,
which is achieved by standardizing both time series (through subtracting the mean and dividing by the standard deviation).
This operation changes the hydrographs in terms of scale, but not in terms of shape and is similar to common methods such



as cluster analysis. It is also used in other hydrological studies employing DTW (Ouyang et al., 2010; Dupas et al., 2015). A property of standardization is that the obtained distances between two standardized time series can become equally spaced in periods without peaks, which makes the method unsuitable for DTW in these circumstances. This was prevented by limiting the analysis to the nine highest discharge peaks. The Derivative Dynamic Time Warping (DDTW) method can provide a solution to the need of standardization (Keogh and Pazzani, 2001), but the minor tidal influence at the lowland confluence makes DDTW not applicable in this case.

The cumulative distance, $r(i,j)$, is the sum of the distance $d(i,j)$ of the current element and the minimum of the cumulative distances of the surrounding elements:

$$r(i,j) = d(x_i, y_j) + \min\{r_{i-1,j-1}, r_{i-1,j}, r_{i,j-1}\}. \tag{2}$$

The warping path, $W$, is a matrix that maps $x$ to $y$ with the lowest cumulative distance and basically reconstructs the $i's$ and $j's$ of the right hand-side of the plus in Equation 2. The length of the path is not necessarily equal to the length of $x$ or $y$ due to the dynamic time lags, where $i$ is not equal to $j$. Hence,

$$W_{l,1} = w_{1,i}, w_{2,i}, ..., w_{l,i}, ..., w_{L,i}, \tag{3}$$

$$W_{l,2} = w_{1,j}, w_{2,j}, ..., w_{l,j}, ..., w_{L,j}, \tag{4}$$

where $W_{l,1}$ and $W_{l,2}$ are the $i's$ and $j's$, respectively, of the lowest cumulative distances with lengths, $L$, of $\max(m,n) \leq L < m + n$. The warping path is subject to the following constraints related to the dependency on the cumulative distance:

1. Boundary conditions: requires the warping path to start in the top left and end in the bottom right of the matrix (Figure 4);

2. Continuity: restricts the allowable step size not to be greater than 1 relative to the previous $i$ and $j$, thus moving horizontally, vertically or diagonally in time;

3. Monotonicity: forces the points in $W$ not to go back in time.

Considering discharge wave propagation, the time lags between stations cannot be infinite. Therefore, warping paths greater than one week are not allowed, but this warping path restriction can be increased for larger catchments. A result of the warping path is shown in Fig. 4. The time lag was calculated according to:

$$t_l = W_{l,1} - W_{l,2}. \tag{5}$$





This time lag was not constant during the analyzed discharge wave, for example due to diffusion. When a diffusive discharge wave is compared with a non-diffusive discharge wave, the slope of the rising and falling limbs differ between the two waves. This difference results in a variation of the time lag along the rising and falling limbs of the discharge waves, regardless of the time lag influenced by advection. The modal time lag was interpreted as the value for the time lag associated with the discharge peak, as the modal time lag is not influenced by diffusion. The duration of the discharge peaks was defined as the time the

discharge is above the 5 % highest discharges over the fifteen analyzed years.

## 2.2 Study Area and Data

### 2.2.1 River Meuse

The Meuse drains an area of 33,000 km$^2$ between northern France and the Netherlands (Figure 2) and experiences a temperate climate. The Meuse is mainly rain-fed and has an average annual discharge of 350 m$^3$ s$^{-1}$. The rain-fed flow regime is erratic

and the catchment has different geological and orographic settings, causing different response times and precipitation patterns within the Meuse catchment area (Berger, 1992; De Wit et al., 2007; Leander et al., 2005). The Meuse can be divided into three parts: Meuse Lorraine, Ardennes Meuse and the lowland Meuse (see De Wit et al., 2007). Meuse Lorraine has mild valley slopes and lies between two ridges. It therefore responds temperately to precipitation, which is partly retained in reservoirs. The Ardennes Meuse consists mainly of hard rock, has steep valley slopes, and as such responds flashy, besides some retention

in reservoirs. The lowland Meuse has very mild slopes and lies in deep alluvial deposits and therefore responds slowly to precipitation. The largest amount of precipitation falls in the Ardennes and therefore contributes most to the discharge (Leander et al., 2005; Rakovec et al., 2012).

### 2.2.2 Tributaries Dommel and Aa

The Dommel and Aa catchments represent 5 % of the Meuse catchment and flow from the Belgian Kempen region to the

Dutch city of 's Hertogenbosch (Figure 2). The average discharges of the Dommel and the Aa are 14 m$^3$ s$^{-1}$ and 8 m$^3$ s$^{-1}$, respectively, and their gradients are ∼ 75 m per 100 km. The Dommel and Aa are rain-fed and, due to the high groundwater levels, have a flashy character despite the rivers' deep alluvial basins. The Aa has been straightened and canalized in the past hundred years. The Dommel has mostly kept its natural plan form. Downstream of the confluence of the Dommel and Aa, the river is called Dieze. When the water level in the Dieze is less than 5 centimetres higher than the Meuse, the weir in the Dieze

is closed and the water discharges through the Drongelse Canal, which has a capacity of 100 m$^3$ s$^{-1}$. The aim of the river training is to prevent the Meuse from flooding areas in the Dommel and Aa catchments. Retention areas of 8.60 million m$^3$ are used when the discharge capacity of the Drongelse Canal is not sufficient in discharging the Dommel and Aa when the weir to the Meuse is closed. The weir in the Dieze was closed only for a few hours in the focus period 1999–2015, namely during the January 2011 event.



### 2.2.3 Discharge and Precipitation Data


In this study, we used hourly measured water levels and discharges of the Meuse at Megen (MU) and of the Dommel (DD) and Aa (AD) between 1999 and 2015 (Figure 1). Flow velocities were measured with a FLOW 2000 measuring device and were converted to discharges using a known cross sectional area. The water level differences were computed by subtracting MD from MU, and DA from DD or AD (Figure 1). The water levels in this area are controlled by weirs and dikes due to the high flood risk, but the position of the gauging stations were chosen such that the water levels would be least influenced by the weirs.

In addition to discharge and water level data, a data set of daily precipitation for Europe was used (Haylock et al., 2008). Precipitation sums were computed for the combined Dommel and Aa catchment areas and the Ardennes catchment area. The latter catchment has the largest contribution to the discharge of the Meuse (illustrated by the green boxes in Fig. 2). The highest discharge and precipitation peaks will generally have the largest flood implications in case of simultaneous occurrence. The nine highest discharge peaks at Megen (MU) were analyzed (Figure 3). High evaporation rates in summer cause a strong seasonal discharge cycle. The highest precipitation peaks in summer will therefore be stored in the soil or evaporated, and will not result in discharge peaks. As a result, the timing of the nine highest discharge peaks does not match with the nine highest precipitation peaks in the Meuse and Dommel.

### 2.3 Response Times and Time Lags

Simultaneous occurrence of discharge waves is caused by a combination of similar precipitation and discharge patterns in two catchments (Betterle et al., 2017). We first analyze the rainfall-discharge response times for the nearest gauging stations upstream of the confluences subject to study, indicated with DD, AD and MU in Fig. 1. For these stations, we calculate the time lags between peaks in rainfall and discharge using the dynamic time warping method (DTW). The time lags are calculated in days and hours for ,respectively, the rainfall and discharge.

Subsequently, we analyse the travel time needed for a discharge peak to move from the gauging station to the confluence, based on the celerity of the flood wave. In order to be able to determine the time lag between a main river and a tributary at a confluence, the celerity from the nearest gauging station to the confluence is determined and added to the timing of the measured discharge peaks. The celerity can be approximated by:

$$t_c = \frac{s}{\sqrt{g \cdot h}},\qquad(6)$$

where $t_c$ (s) is the travel time of a discharge peak from gauging station to confluence, $s$ (m) the distance from gauging station to confluence, $g$ (m s$^{-2}$) the gravitational acceleration and $h$ (m) the mean water depth during the discharge peak. The average travel time from the gauging station to the confluence for the Meuse is 2 hours. It is negligible for both the Dommel and the Aa. Hence, in order to calculate the degree of simultaneous occurrence of discharge peaks at the confluence, 2 hours are subtracted from the time lag between the main river and tributary determined from the gauging stations.

30





Together, these two time lags serve to understand the meteorological and hydraulic conditions leading to the discharge events. In addition, the time lags between the main river and its tributaries were determined, again based on DTW.

## 3   Results

Rainfall in the Meuse river basin is mainly concentrated in the Ardennes and surrounding area as a result of orographic
5   effects (Figure 5). Even though the combined Dommel/Aa catchment and the Ardennes Meuse catchment are separated by
150 kilometres, daily precipitation patterns over the two catchments are spatially correlated (Figure 6). Figure6 shows
the daily precipitation sum in the Meuse and the Dommel catchment areas over the past 45 years. As result of the multiple
measurements without precipitation, the authors have transformed the measurements into a Gumbel distribution to focus on the
high precipitation events. The scatter plot shows that the simultaneous occurrence of the high precipitation events in both the
Meuse and the Dommel catchment areas occur 2.9% over the past 45 years in relation to 5% and 0.25% in case of, respectively,
complete correlation and complete randomness. The daily precipitation sum of the Meuse and the Dommel is not completely
correlated due to travel times and increase or decrease of rainfall events. The lines in Figure 6 indicate that most precipitation
events do not consist of a single day event and therefore a combination of multi-day and heavy precipitation event results in
high discharge events. The highest discharge events are caused by multi-day precipitation events or by a series of precipitation
events with an interval of a couple of days.

The average response times between precipitation and discharge at the three gauging station close to the junctions subject
to study are 3 days for MU (Figure 5), 2 days for DD and 1 day for AD, with standard deviations of approximately half
the average response times. These standard deviations emphasize the variability of the events. The response times suggest that
precipitation events occurring within a two-day interval may lead to a higher probability of simultaneous occurrence of the
maxima of discharge waves. Inundations with societal impacts are reported for the discharge events of 2003, 2010 and 2011 in
Wallonia (Belgium) and in the southern regions of the Netherlands. The computed response time for the severe precipitation
event of 2003 to a discharge peak is negative, because the discharge measuring device failed and consequently leads to an
incomplete discharge wave. The high discharge events show that the response time from precipitation to discharge not only
depends on the intensity and duration of the precipitation event, but also on wave attenuation (Woltemade and Potter, 1994;
Turner-Gillespie et al., 2003; Sholtes and Doyle, 2011) and initial conditions of the catchment before the discharge peak, such
as antecedent soil moisture (Figure 3).

Our results show that the precipitation patterns for the Dommel/Aa and Meuse catchment areas are correlated (Figure 6),
which underlines the importance of the time lags between the discharge peaks of the Meuse and the Dommel/Aa to assess the
potential for simultaneous occurrence of discharge waves. The discharge waves in the Dommel and Aa arrive at the confluence
3.2 and 2.7 days prior to the discharge wave in the Meuse, respectively (Figure 7). The time lag of the Aa is thus half a day
smaller, likely because of the shorter drainage length and the higher degree of canalization of the channels. The corresponding
standard deviations are 16 hours (0.67 days) for the Dommel and 28 hours (1.17 days) for the Aa. Note that the discharge
of the Aa has been measured only since 2004, which explains the absence of time lags prior to 2004. We conclude that the





average time lag between the arrival of discharge peaks from the main river and the tributaries at the confluence is larger

than the average response time of discharge at the monitoring stations to precipitation events. Thus, when processes of runoff generation are fast, simultaneous occurrence of discharge peaks is unlikely.

Although discharge peaks may not coincide, the question remains whether the time lag between the discharge waves is large enough to prevent simultaneous occurrence of high discharges. Figure 8 shows the precipitation of the Meuse and the tributaries Dommel and Aa, as well as the standardized discharge waves. The colored discharge waves are the periods during

which the standardized discharge exceeds the 95th percentile of the time series. Unfortunately, some data gaps (caused by measurement errors) are present in the time series during discharge waves. Despite this, it is evident that all nine highest discharge waves show color overlap and thus coincide.

The fact that simultaneous occurrence of discharge waves takes place for all analyzed extreme events can be explained from the relatively long durations of the discharge waves. The duration of the discharge waves is 9 days on average at MU and even

13 days for the three highest discharge waves (Figure 7). The average time lag between the discharge waves of 3 days is therefore too small to prevent simultaneous occurrence. Figure 9 shows the discharges in the Meuse and Dommel upstream of the confluence over the analysed 15 years , with and without applying a 3-day time shift corresponding to the left and right panels (respectively). Similar to the precipitation sum, the discharges are transformed into a Gumbel distribution to emphasize the high discharges over the low discharges. The Meuse and Dommel discharge are in 2.5% of the time both higher than 5%

of their discharge, which is in between complete correlation and complete randomness. The Meuse and Dommel discharges are not completely correlated as consequence of the time lag between Meuse and Dommel discharge. The application of the time shift increases however the correlation for the highest discharges only slightly, which is apparent from the increase of data points in the top right quadrant from 2.5% to 2.7%. The lines in Figure 9 show that, besides the time shift, most of the nine events move through all quadrants, which implies that the scatter is due to the different peak duration length by for example,

spatial rainfall distribution and size of catchment.

From a water management perspective, it is relevant to establish the degree to which simultaneous occurrence of discharge waves affects extreme water levels. During high discharge events the water level difference in the Meuse between MU and MD (Figure 1) increases by about 1.5 m over a distance of 40 km (Figure 10). Afterwards, it decreases to the water level difference typical for normal discharges (0.5 to 1 m). The water level differences in the Dommel (between DD and DA) and Aa

(between AD and DA), on the other hand, decrease from about 1.5 m to 0.5 m and in some cases the difference even vanishes during peak discharges, and increases afterwards over a distance of 4 and 6 km, respectively (Figure 10). Hence, the water level differences decrease during a discharge event in the tributaries, reflecting backwater effects of the Meuse on the tributaries also visible in the the corresponding stage-discharge relation (Figure 11). The backwater height can reach 1.5 m over 4 km from the confluence in the Dommel and Aa tributaries.



## 4 Discussion

The exact time lag between discharge waves in the main river and the tributary is shown to be less relevant than the duration and magnitude of the discharge waves. The impacts of simultaneous occurrence of discharge waves depend strongly on the detailed hydrograph of the discharge wave, which may have a composite character. The probability of simultaneous occurrence of discharge peaks, as referred to in existing studies (De Wit et al., 2007; Vorogushyn and Merz, 2013), becomes less relevant when the duration of the discharge waves becomes large compared to the time-lag between the arrival of discharge waves from branches joining at a confluence. The discharge magnitude before or after the peak can be relatively high, and can lead to backwater effects in the tributary similar to those generated during a true event of wave peak coincidence. The choice of an appropriate discharge threshold is critical in the analysis, because it determines the portion of the discharge wave taken into consideration. The threshold here employed to isolate the 9 extreme discharge events does not influence the results of the DTW method, which shows that DTW can be considered a robust tool when analysing the interaction between discharge waves. Catchment properties and climate characteristics are known to determine the duration of discharge peaks (Gaál et al., 2012). This study adds main stream-tributary interaction as an important factor influencing the local duration of a discharge wave in lowland areas.

Although the approach to studying discharge dynamics presented herein is generic, the results of the application can be argued to be not representative for lowland areas in general. The climatological precipitation maxima in the Meuse catchment are concentrated relatively close to the sea due to orographic effects of the Ardennes, which may not be expected in many other lowland areas. However, if most precipitation would be concentrated further upstream in the Meuse basin, the difference in travel time would merely increase by about two days (Figure 12). Two days extra travel time would still result in simultaneous occurrence of discharge peaks at the confluence for most of the analyzed discharge waves.

The common practice of determining the discharge using stage-discharge relations is not applicable near confluences, because backwater effects apparent as hysteresis in a rating curve cannot be readily accounted for (Figure 11) (Hidayat et al., 2011, 2016). During the highest discharge waves, backwater variation shows to be most severe. For this reason, the exceedance levels of the upstream stations of the confluence cannot directly be projected to locations closer to the confluence. Establishing the exceedance levels and stage-discharge relations for regions near a confluence therefore presents challenges for water management and introducing spatial variation in flood risk.

The aim of the current Dutch water policy is to retain water in small catchments contributing to the main rivers, mainly to prevent drought and to improve water quality. From a flood risk perspective, water should preferably be retained for the duration of the discharge peak in the main river. An existing concern is that the current practice of water retention in the Dommel and Aa catchments increases flood risk, by enhancing the probability of coinciding flood waves. Our analysis puts the importance of the relative timing of flood waves in perspective. The average duration of extreme discharge events is 9 days in the study area, and it is not so relevant when in this period the peaks in discharge of the Dommel and Aa occur. A significant reduction in flood risk would only be achieved when the water is retained for a period covering multiple rainfall events, which is far from the present-day situation.





## 5 Conclusions

Extreme discharge events at the confluence of the River Meuse and two joining lowland tributaries are studied, introducing a new method of analysis based on dynamic time warping. The method offers robust means of tracing individual discharge waves in discharge time series collected throughout a catchment. The study shows that the precipitation patterns in the catchment areas are spatially correlated. Spatial correlation of the precipitation patterns is a prerequisite for simultaneous occurrence at the confluence. From a comparison of the nine highest discharge waves in the main stream and the joining lowland tributaries it follows that the exact timing of the discharge peaks and the probability of simultaneous occurrence of discharge peaks are

little relevant to flood risk. The duration of the discharge wave in the main stream is large compared to the time lags between discharge peaking in the main channel and the tributaries. Initial catchment characteristics produces ambiguous discharge responses to precipitation, such that the timing of duration and magnitude of the discharge peak relative to the precipitation is variable. When discharge waves coincide, the water level difference in the Meuse increases and the water level differences in the tributaries Dommel and Aa decrease. The decrease of water level differences indicates backwater effects in the tributaries

due to simultaneous occurrence. The backwater height can increase to 1.5 m over 4 km from the confluence in the Dommel and Aa rivers. A public belief is that rapid drainage in a lowland tributary will reduce flood risk, because it diminishes the likelihood of coincident discharge peaks in the main stream and the tributary. In addition, there is a concern that measures of water retention, for example to prevent drought and to improve water quality, will increase flood risk. Our analysis puts this concern into perspective, as a systematic retention in the order of days will only marginally affect peak water levels.

*Code and data availability.*   The DTW code can be requested from the corresponding author and the data can be requested at the mentioned organizations:

   – Dutch water level and discharge data are measured by Rijkswaterstaat, which are available via https://www.rijkswaterstaat.nl/water/waterdata-en-waterberichtgeving/waterdata/index.aspx

   – Walloon water level and discharge data are measured by Service Publique de Wallonie (SPW), which are available via http://voies-

hydrauliques.wallonie.be/opencms/opencms/fr/hydro/Archive/annuaires/index.html

   – French water level and discharge data are measured by Direction Générale de la Prévention des Risques/Service des Risques Naturels et Hydrauliques (DGPR/SRNH), which are available via http://www.hydro.eaufrance.fr/

   – Dommel water level and discharge data are measured by "De Dommel" water board, which are available via the contact person Michelle Berg

– Aa water level and discharge data are measured by "Aa en Maas" water board, which are available via the contact person Pim van Santen

   – Rainfall data are collected from the E-OBS dataset, which are available via http://www.ecad.eu

*Author contributions.*   TEXT





*Competing interests.* The authors declare that they have no conflict of interest

*Acknowledgements.* This research is part of the research programme RiverCare, supported by the Dutch Technology Foundation STW (currently TTW), which is part of the Netherlands Organization for Scientific Research (NWO), and which is partly funded by the Ministry of Economic Affairs under grant number P12-14 (Perspective Programme). The authors furthermore would like to thank Dutch Ministry of In-

5  frastructure and Environment (Rijkswaterstaat); Service Publique de Wallonie, Direction générale opérationnelle de la Mobilité et des Voies hydrauliques, Département des Etudes et de l'Appui à la Gestion, Direction de la Gestion hydrologique intégrée; HYDRO Bank MEEDDAT (Direction Générale de la Prévention des Risques/Service des Risques Naturels et Hydrauliques) / DGPR / SRNH ; Dutch water boards "De Dommel" and "Aa en Maas" in collaboration with STOWA, Dutch Foundation of Applied Water Research, for providing discharge and water level data. In addition, the authors acknowledge the E-OBS dataset from the EU-FP6 project ENSEMBLES (http://ensembles-

eu.metoffice.com) and the data providers in the ECA&D project (http://www.ecad.eu). Data can be requested at the aforementioned organizations. The authors furthermore would like to thank Michelle Berg for organising a meeting with the water managers to evaluate the problems and together with Pim van Santen for helping out with the datasets. The authors also would like to thank their colleagues Lieke Melsen and Claudia Brauer for reading the first drafts.



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





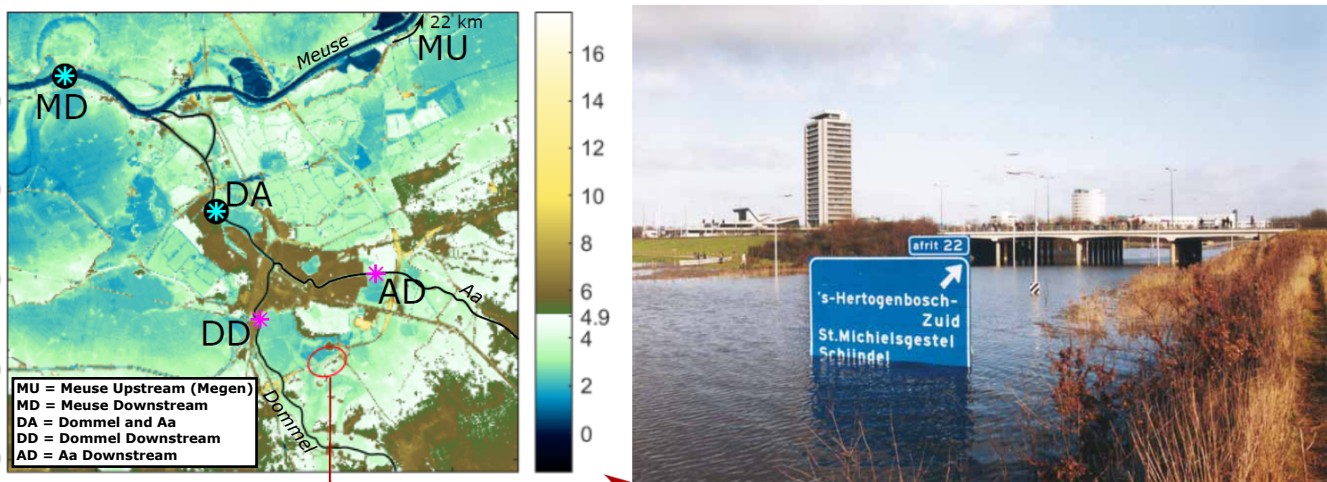

**Figure 1.** Flood potential near the confluence of the Meuse and the tributaries Dommel and Aa. Left panel: DEM of the area around the Dutch city of 's Hertogenbosch. The green and blue colors indicate the flood area to be expected without dikes, based on a maximum water level in the Meuse of 4.9 m above MSL that was reached in January 1995 (also the maximum protected flood level). The magenta asterisks indicate gauge stations with discharge and water level data and the cyan asterisks indicate stations with only water level data. The red circle indicates the flooded area of the European highway E25 (right panel; IJpelaar et al., 2009).





**Figure 2.** Map of the Meuse, Dommel and Aa catchments. The locations of the main gauging stations in the Meuse are shown. Color indicates elevation, and green boxes indicate the areas over which the precipitation is assumed representative for the Meuse and Dommel/Aa catchments, respectively. The white box indicates the area of Figure 1





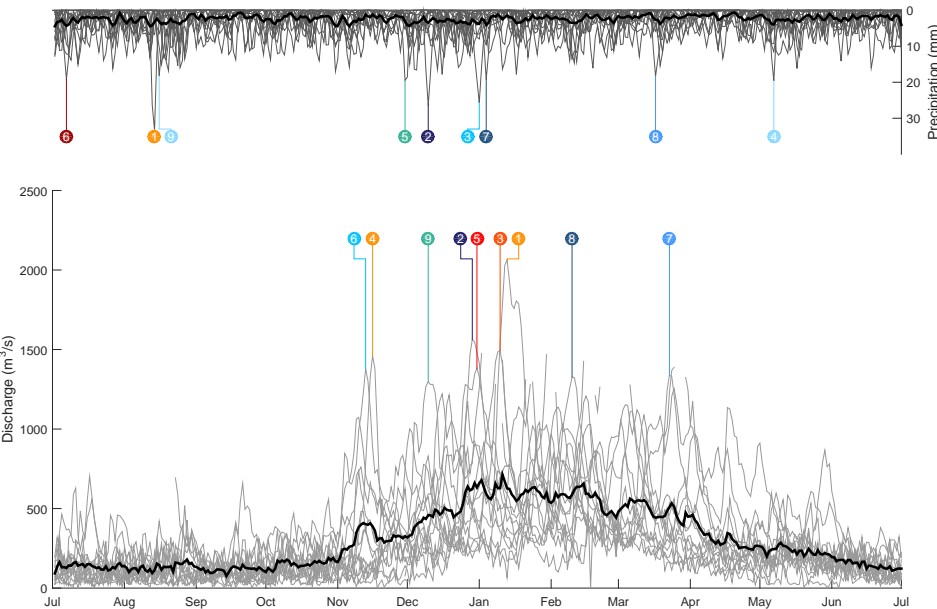

**Figure 3.** Precipitation and discharge extremes for the Meuse. The grey lines show measurements and the black line shows the average value over the measuring period. The colors of the peaks indicate the year and the numbers the decreasing order of the maximum discharges. Note the strong seasonality in discharge, with peaks only occurring in the period November–March, whereas precipitation events occur year-round.



**Figure 4.** Principle of Dynamic Time Warping. The original time series were used for the cumulative distance matrix, which shows the cumulative distance between points in the time series on the $x$-axis and $y$-axis. The line shows the most efficient warping path. The warping path was used to construct the warping signals. This figure shows the DTW method for daily discharge time series at Megen and Venlo (see Figure 2). The result is a time lag of 2 days for this flood event.







**Figure 5.** Distribution of the 5-day precipitation sum preceding the discharge peaks at Megen (MU). The red circle represents the location and the amount of the highest daily precipitation sum (mm). The numbers in the boxes show the time delay between the day with the largest precipitation sum and the discharge peak at MU (d). The flood peaks are arranged from highest flood peak (top left) to lowest flood peak of the nine events (bottom right)





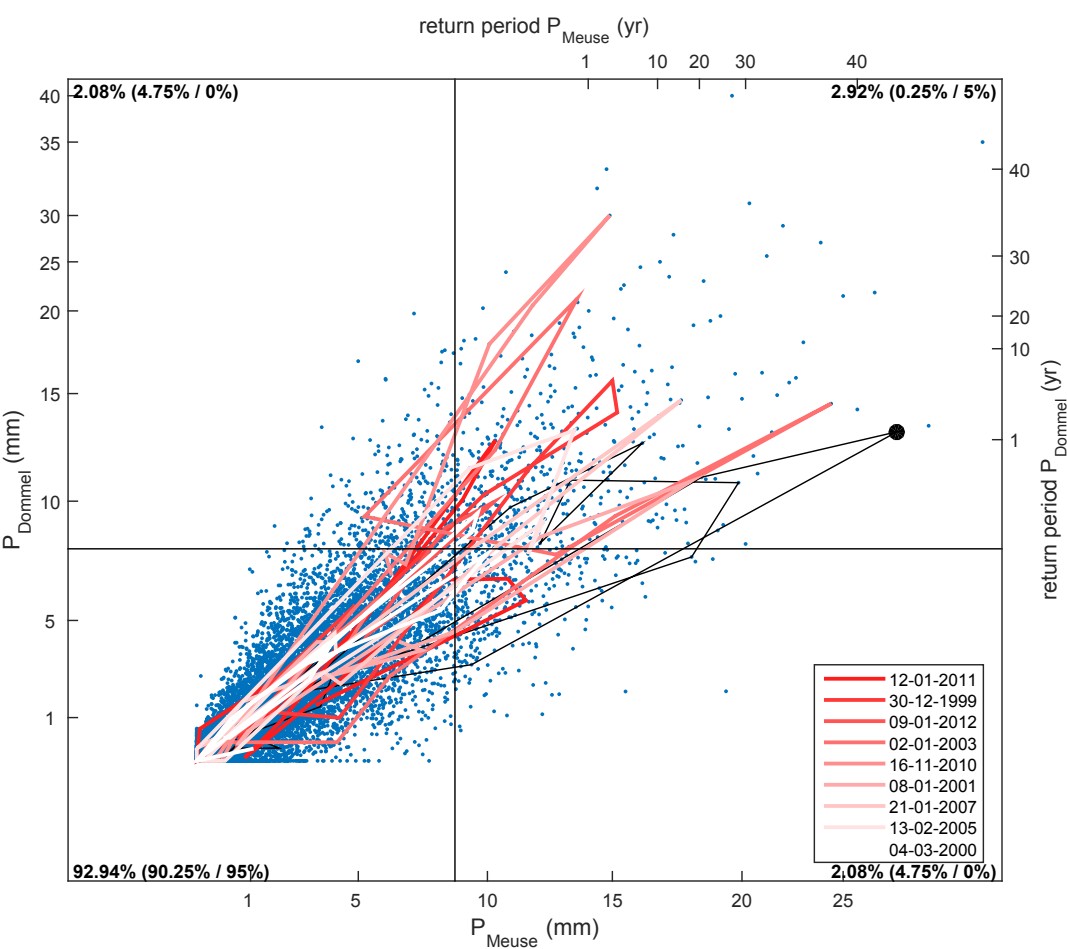

**Figure 6.** Gumbel distribution of the precipitation in the combined Dommel and Aa catchments and the Ardennes from 1968 to 2015. The red lines shows the precipitation a week prior and a week after the analyzed discharge events in Megen, and the black line shows the precipitation two-weeks prior to the flood event in 1995. The figure is divided in quadrants in such a way that 95 % of precipitation values for the Meuse are located at the left-side of the vertical line and that 95 % of precipitation values at the Dommel are located below the horizontal line. The numbers in the corners successively indicate the percentage of samples in each quadrant and the percentages that would result from complete randomness and exact (1-to-1) correlation. The latter two are in brackets.





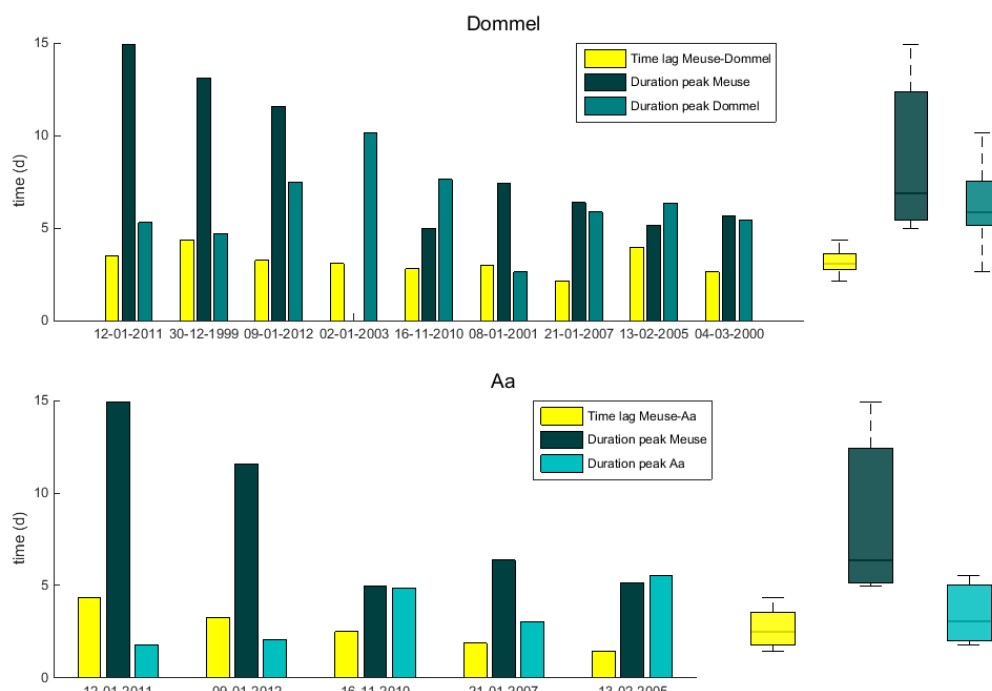

**Figure 7.** Duration of the discharge peaks in the Meuse (MU), the Dommel (DD) and the Aa (AD) rivers and the time lag at the confluence between the Meuse and the Dommel and between the Meuse and the Aa. Box plots show the median, the 25th and the 75th percentiles and the range of the durations and time lags.

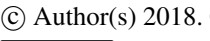

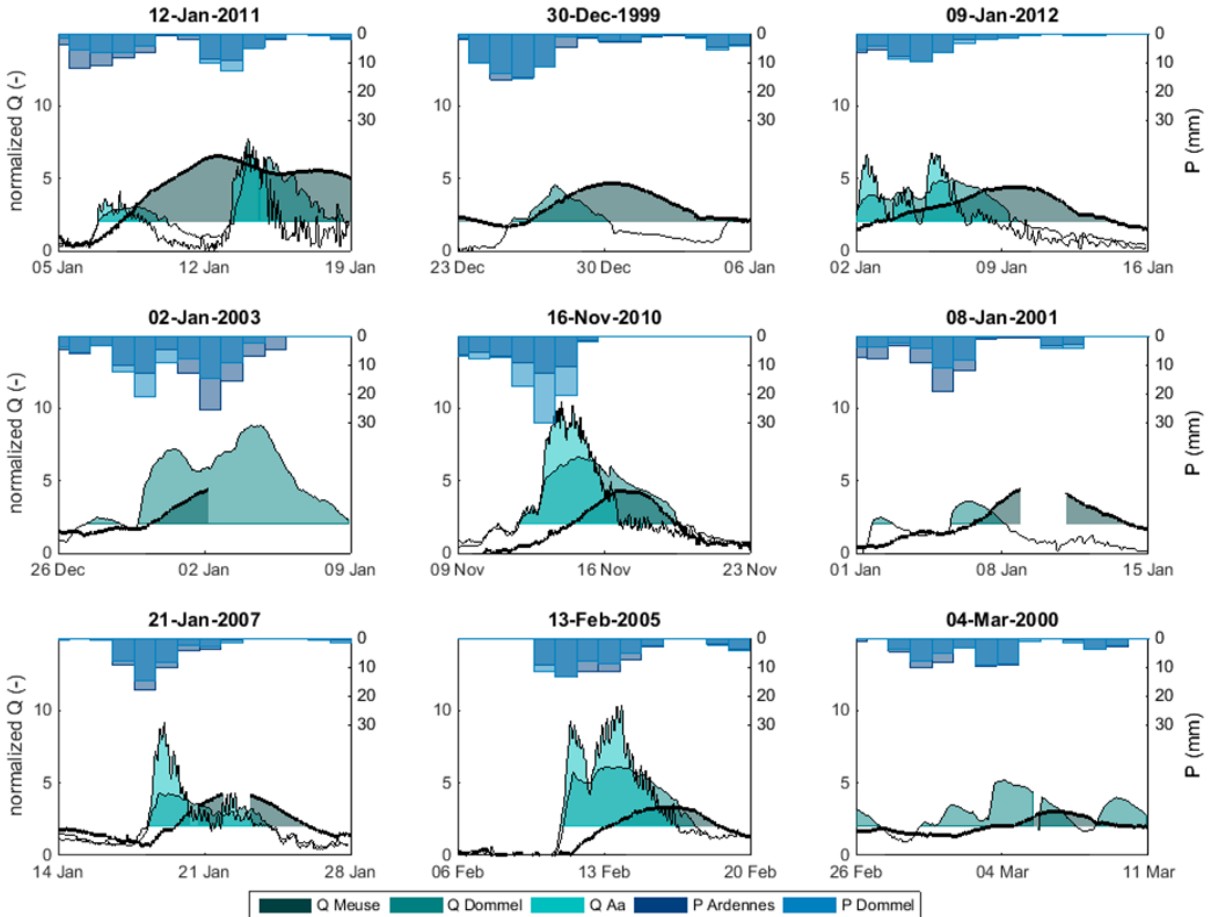

**Figure 8.** Standardized/normalized hydrographs of the Meuse, Dommel and Aa rivers for the nine highest discharge peaks (Figure 3). The colors indicate discharge exceeding the 95th percentile for the Meuse river (dark green), the Dommel river (lighter green) and the Aa river (light turquoise). The overlap between the colors is indicative of simultaneous occurrence. The precipitation in the catchment areas is indicated by dark blue for the Ardennes catchment and lighter blue for the Dommel/Aa catchment.





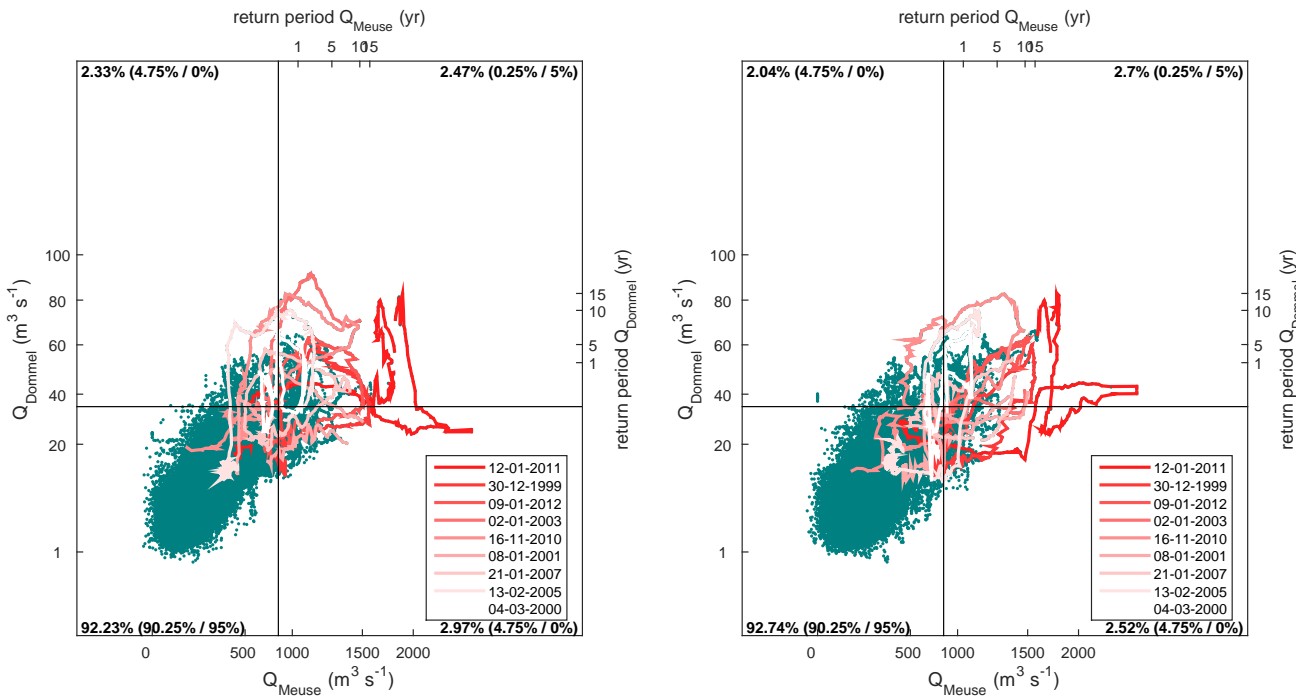

**Figure 9.** Gumbel distribution of the discharge of the Dommel and the Meuse from 1996 to 2015. The red lines show the discharge a week prior and a week after the analysed discharge peaks at Megen and follow a clockwise hysteresis. The discharge of the flood event of 1995 was 2.825 m$^3$ s$^{-1}$ for the Meuse and 100 m$^3$ s$^{-1}$ for the Dommel. The figure is divided in quadrants in a way such that 95 % of discharge values at the Meuse are located at the left-side of the vertical line and that 95 % of discharge values at the Dommel are located below the horizontal line. The numbers in the corners successively indicate the percentage of samples in each quadrant and the percentages that would result from complete randomness and exact (1-to-1) correlation. The latter two are in brackets.





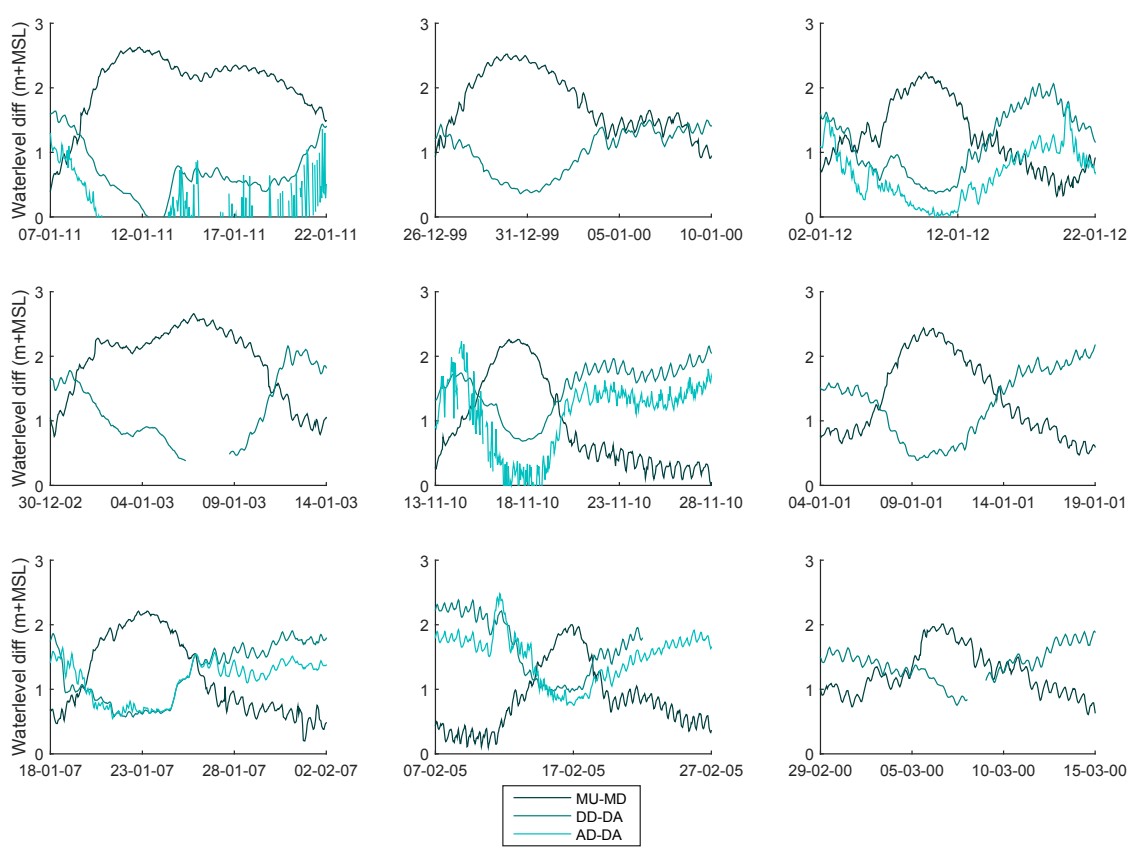

**Figure 10.** Water level differences of the Meuse, Dommel and Aa rivers during the nine highest discharge peaks in the Meuse. Note that the water level difference of the Meuse increases, while the differences of the Dommel and Aa decrease during the discharge peak.





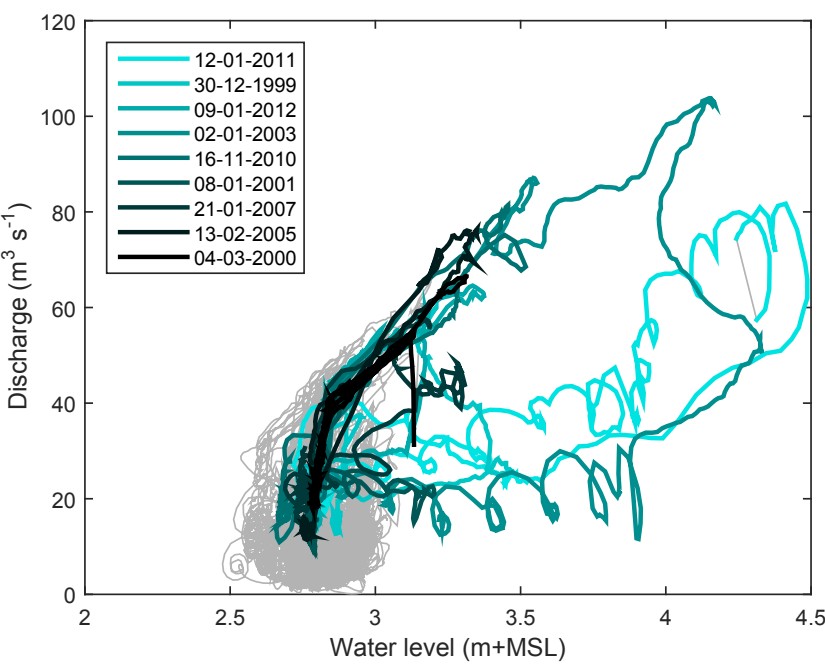

**Figure 11.** The stage-discharge relation of the Dommel river at measuring station DD (Figure 1). Note that stage and discharge have been measured independently, and that strong non-uniqueness and hysteresis effects can be seen.





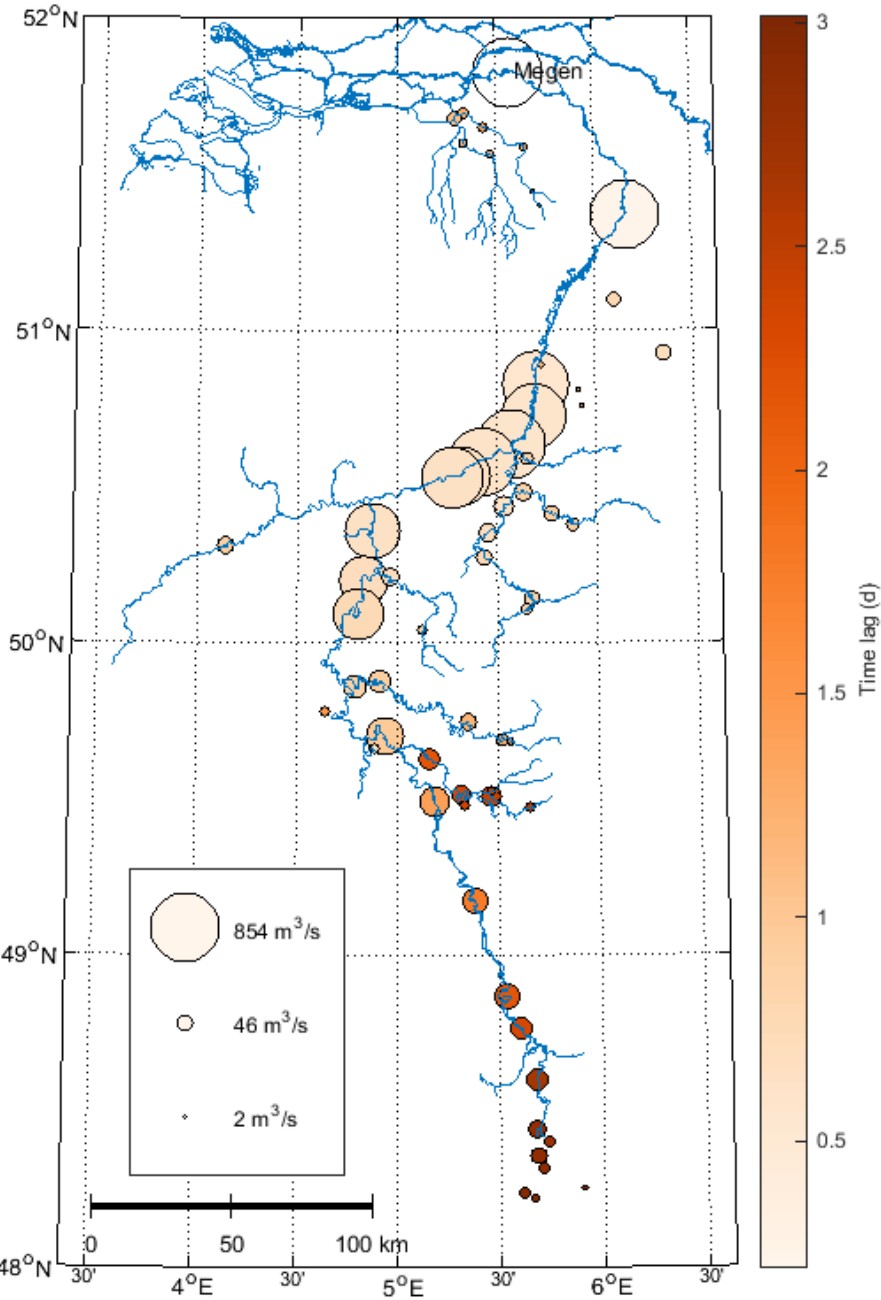

**Figure 12.** Distribution of the average travel time of the nine highest discharge waves in the period 1999–2015, between various gauging stations along the Meuse and the confluence subject to study. The circle size indicates the 95th percentile discharge of the gauging station.