# Peer review of "Anatomy of simultaneous flood peaks at a lowland confluence"

_Hydrology and Earth System Sciences, 2018_

## Referee Comment (RC1) · P. Bates (Referee) · 10 May 2018

This paper looks at the probability of simultaneous flood peaks occurring at river confluences in lowland systems. This is a relatively under-studied topic, yet the impact of network topology and hydrology on the generation of flood extremes is potentially significant and this paper represents an important contribution. Its subject matter is therefore appropriate for HESS. The work is also well executed and well written, and with minor modification to respond to the points detailed below I think the paper can be published.

1. P1, Line 10. The point being made here is that discharge peaks are very likely to coincide at these confluences because of the long duration of flood events in the

Meuse, Dommel and Aa basins. This is a key finding of the paper, but I wonder if this is perhaps as surprising as the paper implies? It is well understood that large river basins (say >100,000 km2) typically have a single annual flood pulse, and this is precisely because flood waves from joining tributaries coincide along the main stem to create a single large wave. In these large basins flood wave duration is typically always long enough that tributary waves can join with the main stem flood pulse. Even the Congo's bi-modal annual flood peak, which is an obvious exception to the 'large basin annual flood pulse' rule, is a product of these effects. My question here is 'at what basin scale does it become surprising that this effect occurs'? Clearly, whether flood waves coincide at tributaries is a product of rainfall patterns over the basin, geology and the network topology, but, from the empirical evidence, at some scale flood wave duration and travel time become sufficiently large that the combination of tributary and main stem flood waves into a larger synchronous pulse becomes almost certain. Should we therefore be surprised to observe coincident waves in a 33,000 km2 lowland catchment? It would be great if the authors could somewhere in the paper add a more nuanced discussion of these effects.

2. P1, Line 20. The point about the highway is repeated here from line 18. It would read better if the repetition was removed.

3. P2, line 12. Same point as (1) above. Does not the empirical evidence from large basins suggest that the counter-hypothesis, namely that superimposition of flood waves is actually to be expected in basins of the scale of the Meuse, is more appropriate?

4. P3, line 2. Isn't the likely response of water levels to simultaneous occurrence of flood peaks already well understood based on standard hydraulic theory? Higher main stem water levels will create significant backwater effects as your analysis shows, but this is well known from other large rivers (see for example Bob Meade's classic paper on backwater effects on the Amazon: Meade, R.H., Rayol, J.M., Da Conceicão, S.C. et al. Environ. Geol. Water Sci (1991) 18: 105. https://doi.org/10.1007/BF01704664).

[Figure]

5. P9, line 13. But the empirical evidence suggests that flood waves quite commonly superimpose in large basins, so whilst other basins may have different climatology the end result seems to be much the same.

6. P9, lines 18 and 19. There are some typos here which make the text on these two lines difficult to follow. Can you edit please?

7. P9, line 22. Should be 'introduces'.

8. P9, line 22. Perhaps should note that this is not the only cause of spatial variation in flood risk.

I hope these comments are useful.

Paul Bates, University of Bristol

---

## Referee Comment (RC2) · Anonymous Referee #2 · 2 Jul 2018

I read the manuscript by Tjitske J. Geertsema et al., entitled "Anatomy of simultaneous flood peaks at a lowland confluence", with a great interest.

Authors propose to analyse the mechanisms leading to simultaneous flood event occurrence at river confluences. I think the work is interesting and represent a good contribution to the research and to the HESS journal. The manuscript is well written and structured. The reading is smooth and easy.

However, I think the manuscript needs some further developments and improvements before it can be published. Please find below some remarks that can help I hope to improve the manuscript.

Major

[Figure]

1. The section related to the DTW method which is central in the method is not clear enough in my opinion. I did my best but unfortunately it seems to me difficult to understand how DTW method works. As mentioned by the authors, this method has not been often applied in hydrology so I think it is important to explain it clearly. The associated figure 4 did not help me to understand the method as well. As a consequence, I warmly recommend the authors to try to reshape and rewrite this section and the associated figure so that a person that does not know how DTW works can follow it step by step.

2. The application of the DTW is not clear enough in my opinion: what are the input data used for instance?

3. The rainfall data should be better presented: are there coming from observation / interpolation? How they are used in the paper is not clear to me: as average value (or sum) over the green boxes in figure 5. I think this is important to understand the concept of time lag between the rainfall and the discharge peak.

4. There are some conclusions of the article that could have been foreseen from the beginning like for instance the influence of the rainfall distribution and the duration of the flood events over the Meuse basin. I would suggest that the authors could already elaborate rapidly on that from the introduction on. Indeed, I found the hypotheses formulated in the introduction sometimes a bit too simplistic in the current version of the manuscript.

Minor

1. Title: I am wondering if the term "anatomy" is really relevant here and then if the title is really representative form the paper?

2. P1 Line 3: Maybe "confluence" would be more appropriate than "merging of rivers".

3. P1 Line 22: "msl" could you please introduce what it stands for.

4. P2 lines 3-13: This view is a little simplistic as it is rather straightforward that the

rainfall spatial distribution is not often uniform (as shown in figure 5).

5. P2 line 24-26: This looks like a conclusion sentence already in the introduction. I would suggest to rephrase it.

6. P3 line 19: I suggest to remind what is the objective of the DTW method first.

7. P3 line 20: It is not clear to me what the "wave traced in discharge time series" means. Could the authors please try to clarify?

8. P 3 line 21: "is" is missing after "this" I think

9. P 6 line 2: could the authors please clarify what is a "FLOW 2000" measuring device: ADCP, current meter? Moreover does this sentence mean that flow velocity is measured continuously?

10. P7 line 5: Could the authors please clarify what they mean by Ardennes Meuse catchment?

11. P7 line 11: The concept of complete randomness is not totally clear to me. Could the authors try to clarify?

12. P7 line 11: sum of daily precipitation: do you mean the sum over the Meuse catchment?

13. P 7 line 16: I think it should be "stations" instead of "station"

14. P 8 line 23: for the sake of clarity, I would suggest to move "over a distance of..." right after "water level difference"

15. P 9 line 3-4: "The discharge magnitude..." I do not really get the point of this sentence. Could the authors please clarify?

16. Figure 1: It is not clear to me what "flood potential" do mean. Could the authors please clarify? Moreover, I do not understand the colour bar (values in m?)? How the expected flooded area is obtained (DEM height values thresholding, if yes I am not

sure this is meaningful)?

---

## Author Comment (AC1) · 13 Jul 2018

Dear editor,

We are very pleased with the constructive reviews and encouragement from both reviewers to proceed towards publication of the work in HESS. When we look at the comments of the reviewers, we conclude that there are no major objections to the methodology used or robustness of the conclusions. In our view, the comments are mainly textual, and below we will outline how we intend to proceed with further improving the manuscript.

From the first review (from Paul Bates) we understand that we need to explain more about the origin of the flood pulse (discussion points 1, 3 and 5). In our view, there is

no annual flood pulse, but there are multiple discharge peaks clustered in the winter season. Paul Bates is right, however, that this depends on the type of catchment. We will discuss this dependency in a counter-hypothesis in the introduction and cover this topic extensively in the discussion. In addition, we will explain the water level response to simultaneous occurrence of discharge waves on the basis of standard hydraulic theory (discussion point 4). The other discussion points of Paul Bates concern smaller textual suggestions, which we will include.

From the second reviewer we understand that we need to explain the DTW method and its application more clearly (major discussion points 1 and 2). We are happy to do this, for which we can elaborate on the input data and better explain figure 4. We intend to revise figure 4 and the text of the method chapter to describe step-by-step the way we used the method. Moreover, we will refer to more literature about the DTW method. In addition, we will better explain the origin of the precipitation data in the text (major discussion point 3) and together with the counter-hypothesis (we formulate, based on comments of the first reviewer), we will elaborate on the precipitation distribution effects (major discussion point 4). The other points of discussion can be addressed by small textual suggestions, which we are happy to incorporate in the manuscript.

We would like to thank again the reviewers for their constructive feedback and for their help to improve the manuscript. We are convinced that an improved version of the manuscript will be of interest to the readers of HESS.

With best regards,

Tjitske Geertsema, on behalf of all authors

---

## Author Response (AR1)

Dear Editor,

Hereby we resubmit our manuscript entitled 'Anatomy of simultaneous flood peaks at a lowland confluence', intended for publication in Hydrology and Earth System Sciences. The manuscript deals with analysis of controls on flooding at a lowland confluence using a unique dataset of independent discharge and water level observations, and application of the Dynamic Time Warping method which has seen few applications in hydrology to date. This research is not only novel and of high relevance, but we also present a methodology that can easily be applied to other regions and/or flood problems.

We would like to thank the reviewers in helping us to improve the significance, application and method of our manuscript. We have taken the comments seriously, as is evidenced by significant textual and conceptual changes. In particular, we have introduced a new figure conceptualizing the main findings for application in other areas (Figure 13) which reflects a long thought-process and internal discussion on the reviews. Also we have expanded the DTW method section. The DTW method has been extended with a paragraph about the application and input parameters. In addition, we have explained the method step-by-step and have introduced more references to Figure 4, which we have adjusted.

In short, we have made multiple adjustments to improve the significance, application and method for readers and we would therefore like to ask to consider this manuscript for publication in HESS.

I'm looking forward to your reaction.

Yours sincerely,

Tjitske Geertsema,

PhD candidate

**Author's response:**

Reviewer 1:

1. P1, Line 10. The point being made here is that discharge peaks are very likely to coincide at these confluences because of the long duration of flood events in the Meuse, Dommel and Aa basins. This is a key finding of the paper, but I wonder if this is perhaps as surprising as the paper implies? It is well understood that large river basins (say >100,000 km2) typically have a single annual flood pulse, and this is precisely because flood waves from joining tributaries coincide along the main stem to create a single large wave. In these large basins flood wave duration is typically always long enough that tributary waves can join with the main stem flood pulse. Even the Congo's bi-modal annual flood peak, which is an obvious exception to the 'large basin annual flood pulse' rule, is a product of these effects. My question here is 'at what basin scale does it become surprising that this effect occurs'? Clearly, whether flood waves coincide at tributaries is a product of rainfall patterns over the basin, geology and the network topology, but, from the empirical evidence, at some scale flood wave duration and travel time become sufficiently large that the combination of tributary and main stem flood waves into a larger synchronous pulse becomes almost certain. Should we therefore be surprised to observe coincident waves in a 33,000 km2 lowland catchment? It would be great if the authors could somewhere in the paper add a more nuanced discussion of these effects.

*The above comment has helped us to put the study in perspective and we would thank the reviewer therefore. We have divided the answer to the comment in three paragraphs namely in (1) an overview of the factors that influence the simultaneous occurrence of discharge peaks, (2) a direct answer to the above comment and (3) the way we have addressed the overview and answer in the revised manuscript.*

*Discharge peaks at the confluence coincide when the time to peak discharge at the confluence and the duration of the discharge wave at the confluence from one catchment area is equal to the time to peak discharge and the duration of the discharge wave from the other catchment area. Discharge waves can thus still coincide if the difference in time to peak discharge at the confluence is large, namely when one or both discharge waves are much longer than the difference in time to peak discharge at the confluence. In contrast, the discharge waves cannot coincide even if the difference in time to peak discharge at the confluence is short, namely when the discharge waves are even shorter in duration at the confluence. Based on a literature review and our analysis, we have described the various factors that influence the time to peak discharge and the duration of discharge waves in Figure 13, where the pluses indicate a positive effect and the minuses indicate a negative effect. It should be noted that the time to peak discharge also influences the duration of discharge waves and that the difference between catchment areas is important to understand simultaneous occurrence of discharge peaks.*

*With the help of figure 13, we will discuss the comment given by the reviewer. The basin size is not included in the figure, because the basin size influence simultaneous occurrence in two ways. Firstly, large basin areas have a high probability of multiple tributaries and thus a longer duration of the discharge wave, which increases the likeliness of the simultaneous occurrence of drainage peaks. This effect is argued by the reviewer. However, large basin areas can also lead to longer time to peak discharge at lowland confluences because of the larger drainage network. The Mississippi river, for example, does not have an annual flood pulse but several smaller flood pulses according to our definition. While the Mississippi river has the fifth to largest basin area in the world. The two largest basin areas of the Amazon and Congo river have indeed annual or bi-annual flood pulses, but we conclude from figure 13 that not only the amount of tributaries contributes to the long duration of drainage waves*

*but also the long duration of precipitation in tropical areas. It can rain for four months in a row in the Amazon catchment, which makes the 4-month discharge wave less surprising.*

*We agree with the reviewer that figure 13 can make a good contribution to the discussion of the paper, both for climate and catchment area characteristics. We have therefore added the new conceptual figure to the discussion chapter.*

2. P1, Line 20. The point about the highway is repeated here from line 18. It would read better if the repetition was removed.

*Thanks for the notification. We have removed the repetition (P1, lines 19-22).*

3. P2, line 12. Same point as (1) above. Does not the empirical evidence from large basins suggest that the counter-hypothesis, namely that superimposition of flood waves is actually to be expected in basins of the scale of the Meuse, is more appropriate?

*We have removed the hypothesis. In the current paragraph we describe the factors influencing simultaneous occurrence ( P2, lines 13-33) and include references to the paper of Bob Meade (see next comment).*

4. P3, line 2. Isn't the likely response of water levels to simultaneous occurrence of flood peaks already well understood based on standard hydraulic theory? Higher main stem water levels will create significant backwater effects as your analysis shows, but this is well known from other large rivers (see for example Bob Meade's classic paper on backwater effects on the Amazon: Meade, R.H., Rayol, J.M., Da Conceicão, S.C. et al. Environ. Geol. Water Sci (1991) 18: 105. https://doi.org/10.1007/BF01704664).

*We agree and have changed the research question (P3, lines 5-6.) and included reference to the paper of Meade in the paragraph describing the involved factors in simultaneous occurrence of discharge peaks.(P2, line 28 and 31)*

5. P9, line 13. But the empirical evidence suggests that flood waves quite commonly superimpose in large basins, so whilst other basins may have different climatology the end result seems to be much the same.

*We agree that a part of the discussion is missing and therefore we rewrote the paragraph in the discussion as discussed at comment 1 (P10, line 26 to P11 line 15).*

6. P9, lines 18 and 19. There are some typos here which make the text on these two lines difficult to follow. Can you edit please?

*Thanks for the notification. We have changed the text (P11, lines 16-18).*

7. P9, line 22. Should be 'introduces'.

*We have changed the text in 'introduces' (P11, line 21).*

8. P9, line 22. Perhaps should note that this is not the only cause of spatial variation in flood risk.

*We included this comment in the text (P11, line 21).*

Major

1. The section related to the DTW method which is central in the method is not clear enough in my opinion. I did my best but unfortunately it seems to me difficult to understand how DTW method works. As mentioned by the authors, this method has not been often applied in hydrology so I think it is important to explain it clearly. The associated figure 4 did not help me to understand the method as well. As a consequence, I warmly recommend the authors to try to reshape and rewrite this section and the associated figure so that a person that does not know how DTW works can follow it step by step.

*We have changed the section about the DTW method by introducing a step-by-step approach. In addition, we have changed figure 4 and refer multiple times to the figure in the text (P3-6).*

2. The application of the DTW is not clear enough in my opinion: what are the input data used for instance?

*We have expanded the first paragraph of the DTW method with detailed description of the application and the input data of the DTW method (P3, line 29 to P4 line 14).*

3. The rainfall data should be better presented: are there coming from observation / interpolation? How they are used in the paper is not clear to me: as average value (or sum) over the green boxes in figure 5. I think this is important to understand the concept of time lag between the rainfall and the discharge peak.

*The presented rainfall data uses an interpolation of observed rainfall data and we have used the data by taking the sum over the green boxes in figure 5. We have elaborate on the precipitation data set in section about discharge and precipitation data (P7, lines 17-22) and we have checked the chapter of results in order to describe the precipitation data correctly (P8, lines 12-25).*

4. There are some conclusions of the article that could have been foreseen from the beginning like for instance the influence of the rainfall distribution and the duration of the flood events over the Meuse basin. I would suggest that the authors could already elaborate rapidly on that from the introduction on. Indeed, I found the hypotheses formulated in the introduction sometimes a bit too simplistic in the current version of the manuscript.

*We have removed the hypothesis to elaborate on the factors that increases the likelihood of simultaneous occurrence of discharge peaks in the introduction (P2, lines 13-33 and see also answers on comments 1 and 3 of reviewer 1).*

Minor

1. Title: I am wondering if the term "anatomy" is really relevant here and then if the title is really representative form the paper?

*We have changed the title in "Simultaneous occurrence of discharge peaks at a lowland confluence"*

2. P1 Line 3: Maybe "confluence" would be more appropriate than "merging of rivers".

*We have changed the text to "in merging rivers at confluences", because referring only to confluences is not precise since confluences can also be present in other than lowland areas. (P1, line 3)*

3. P1 Line 22: "msl" could you please introduce what it stands for.

*We have included the explanation of mean sea level (msl) at P1, line 23*

4. P2 lines 3-13: This view is a little simplistic as it is rather straightforward that the rainfall spatial distribution is not often uniform (as shown in figure 5).

*We removed this whole section from the introduction and included a more elaborate discussion on the factors influencing simultaneous occurrence of discharge peaks at lowland confluences (P2, lines 13-33).*

5. P2 line 24-26: This looks like a conclusion sentence already in the introduction. I would suggest to rephrase it.

*We have rephrased the sentence in "While in rain-fed systems, the variability in the hydrograph shapes of individual peak discharge events can be so large, that changes in the relative timing cannot readily be translated to a change in flood risk." (P2, lines 10-12)*

6. P3 line 19: I suggest to remind what is the objective of the DTW method first.

*The objective of the DTW method is to determine the time lag between rainfall and runoff, and between the nine highest discharge peaks of two catchments. We have described the objective in the first paragraph of the DTW method (P3, line 29 to P4 line 14).*

7. P3 line 20: It is not clear to me what the "wave traced in discharge time series" means. Could the authors please try to clarify?

*We meant the way the wave is defined in the discharge time series. We have rephrased the sentence in "The advantage of DTW is that no assumptions are needed regarding the definition of a wave in discharge time series" (P4, lines 5-6).*

8. P 3 line 21: "is" is missing after "this" I think

*Thank you for the notification. We have included "is" in the sentence (P4, line 6).*

9. P 6 line 2: could the authors please clarify what is a "FLOW 2000" measuring device: ADCP, current meter? Moreover does this sentence mean that flow velocity is measured continuously?

*FLOW 2000 measuring device is acoustic current meter and the flow velocity is measured every 15 minutes. We have included this information in the discharge and precipitation section (P7, lines 12-13).*

10. P7 line 5: Could the authors please clarify what they mean by Ardennes Meuse catchment?

*The Ardennes Meuse catchment is a typo. We meant the Ardennes catchment, which is part of the Meuse basin. We have changed the Ardennes Meuse catchment in the Ardennes catchment, which is the largest of the two green boxes in Figure 2 (P8, lines 12-25).*

11. P7 line 11: The concept of complete randomness is not totally clear to me. Could the authors try to clarify?

*We mean with complete randomness that there is no correlation, so the observations between the two catchments are randomly distributed. In that case 90.25% of the two catchment measurements are located in the lower left quarter, 4.75% in the lower right and upper left quarter and 0.25 % in the upper right quarter. We changed "complete randomness" in "in the case of randomness" (P8, line 21). We prefer randomness above no correlation since no correlation sometimes refers to a certain correlation value.*

12. P7 line 11: sum of daily precipitation: do you mean the sum over the Meuse catchment?

*We indeed mean the sum over the catchment. We have elaborate on the method in the discharge and precipitation data section at P7, lines 17-22.*

13. P 7 line 16: I think it should be "stations" instead of "station"

*Thanks for the notification. We have changed it (P8, line 26).*

14. P 8 line 23: for the sake of clarity, I would suggest to move "over a distance of: : :" right after "water level difference"

*Thanks for the notification. We have changed the sentence (P10, lines 4-5).*

15. P 9 line 3-4: "The discharge magnitude: : :" I do not really get the point of this sentence. Could the authors please clarify?

*We mean the amount of discharge. We are trying to make the point that discharge peak should not be considered as a single peak, but as a wave with a certain shape (duration and height). When only the peak discharge would be used for the analysis of simultaneous occurrence of discharge peaks, the amount of discharge just in front or just after the peak is neglected, while these amount of discharges could even so result in simultaneous occurrence of discharge peaks and thus floodings. For clarity, we have changed "discharge magnitude" in "amount of discharge" (P10, line 18)*

16. Figure 1: It is not clear to me what "flood potential" do mean. Could the authors please clarify? Moreover, I do not understand the colour bar (values in m?)? How the expected flooded area is obtained (DEM height values thresholding, if yes I am not sure this is meaningful)?

*We have changed "flood potential" in "flood proneness", because it is a better term for what we are showing in the figure and "flood area" is a hypothesized area as the flooding never happened. The colour bar is indeed in meters, which we have now indicated in the figure. To clarify our intention with this figure, we have changed the caption ("
[revised manuscript text omitted]

---

## Author Response (AR2)

Dear Editor,

Hereby we send you the final version of our manuscript. We have made the adjustments as suggested by the reviewer, namely:

- The title has been changed back to the "Anatomy of simultaneous flood peaks at a lowland confluence".

- References to the Meuse have been removed from the third paragraph of the introduction.

- The numbering of the figures has been changed

- "Conceptual model" has been changed to "conceptual framework"

- The grid of precipitation data was 0.25 at 0.25 degrees. We corrected the size of grid in the text and changed the reference to the website.

- A reference for equation 5 and a sentence with the reason for the use of the average water depth has been added.

We hope that with these adjustments the manuscript can be accepted for publication in HESS and we are looking forward to your response.

Yours sincerely,

Tjitske Geertsema,

PhD candidate